# Marine biofilms constitute a bank of hidden microbial diversity and functional potential

Weipeng Zhang[1], Wei Ding[1], Yong-Xin Li [1], Chunkit Tam[1], Salim Bougouffa [2], Ruojun Wang[1], Bite Pei[1], Hoyin Chiang[1], Pokman Leung [1], Yanhong Lu[1], Jin Sun[1], He Fu[3], Vladimir B Bajic [2], Hongbin Liu[1], Nicole S. Webster [4,5] & Pei-Yuan Qian[1]

Recent big data analyses have illuminated marine microbial diversity from a global perspective, focusing on planktonic microorganisms. Here, we analyze 2.5 terabases of newly sequenced datasets and the *Tara* Oceans metagenomes to study the diversity of biofilm-forming marine microorganisms. We identify more than 7,300 biofilm-forming 'species' that are undetected in seawater analyses, increasing the known microbial diversity in the oceans by more than 20%, and provide evidence for differentiation across oceanic niches. Generation of a gene distribution profile reveals a functional core across the biofilms, comprised of genes from a variety of microbial phyla that may play roles in stress responses and microbe-microbe interactions. Analysis of 479 genomes reconstructed from the biofilm metagenomes reveals novel biosynthetic gene clusters and CRISPR-Cas systems. Our data highlight the previously underestimated ocean microbial diversity, and allow mining novel microbial lineages and gene resources.

---

[1] Department of Ocean Science and Division of Life Science, Hong Kong University of Science and Technology, Hong Kong, China. [2] Computational Bioscience Research Center, King Abdullah University of Science and Technology, Thuwal 23955, Saudi Arabia. [3] Department of Marine Sciences, University of Georgia, Athens 30602 GA, USA. [4] Australian Institute of Marine Science, St Lucia 4067, QLD, Australia. [5] Australian Centre for Ecogenomics, University of Queensland, St Lucia 4067, QLD, Australia. Correspondence and requests for materials should be addressed to P.-Y.Q. (email: boqianpy@ust.hk)

Microorganisms contribute significantly to the health and resilience of marine ecosystems via their critical roles in biogeochemical cycling[1], their interactions with macroorganisms[2], and their provision of chemical cues that underpin the recruitment of marine invertebrates[3]. Recent advances in sequencing technology have facilitated unprecedented analyses of the ocean microbiome. The *Tara* Oceans project analyzed 7.2 terabases of metagenomic data to illuminate microbial diversity from a global perspective and generated a global ocean microbial reference gene catalog (OM-RGC)[4]. The oceanic microbiome exhibits vertical stratification with community composition primarily dependent on seawater temperature[4]. However, most global oceanic surveys have focused on free-living bacterioplankton, despite many aquatic microorganisms colonizing solid surfaces to form multi-species biofilms[5,6].

Surface association can have numerous ecological advantages including environmental protection, increased access to nutritional resources and enhanced opportunities for interactions with other organisms[6]. Microbial surface association can also have deleterious outcomes such as biofouling, biocorrosion, and antibiotic resistance[6]. Developing biofilms on artificial surfaces deployed in the environment is a useful way to study the behavior of marine biofilms as it allows for the observation of initial microbial attachment, community succession, and microbe-host interactions[7–10]. According to analyses of artificial surface-supported biofilms, the composition of marine microbial biofilms is generally distinct from the surrounding seawater[10] and can be influenced by both location[7] and substratum type[8,10]. In addition, a recent study investigated biofilms formed on marine plastic debris, and pointed out that microbes attached to these plastic fragments are different from the surrounding seawater[11]. However, these insights have been primarily derived from localized analyses, and the extent to which biofilm-forming microbes contribute to global microbial diversity is unknown. Similarly, while gene functions including signal transduction[12], stress response[13], motility inhibition[14], and extracellular matrix dynamics[15] are frequently linked to single-species biofilms, the functional core for complex marine biofilm communities remains to be elucidated.

Ecologically, the principle that 'everything is everywhere, but the environment selects'[16] posits that all microorganisms in the ocean are globally distributed but that in any given environment most of the species are only latently present. This notion has been supported by analysis of microbial communities from environments as varied as the English Channel[17] to Atlantic hydrothermal vents[18].

For the present study, we hypothesized that the oceanic biofilm-forming microbiome contains previously undetected species that are only latently present in seawater, and that these communities maintain a persistent and distinct functional core. To test this, we selected eight locations across the Atlantic, Indian and Pacific Oceans, developed biofilms on different materials, and used deep metagenomic sequencing to compare the community structures with microorganisms in the adjacent seawater and in the entire *Tara* Oceans database.

## Results

**Microbial diversity and structure of marine biofilms.** Locations and materials for biofilm development are shown in Supplementary Figs 1 and 2. In total, 101 biofilm and 24 seawater metagenomes were obtained after microscopic confirmation of the biofilms multicellular structure (Supplementary Fig. 3). In parallel, 67 metagenomes of epipelagic seawater collected from around the world were downloaded from the *Tara* Oceans study[4].

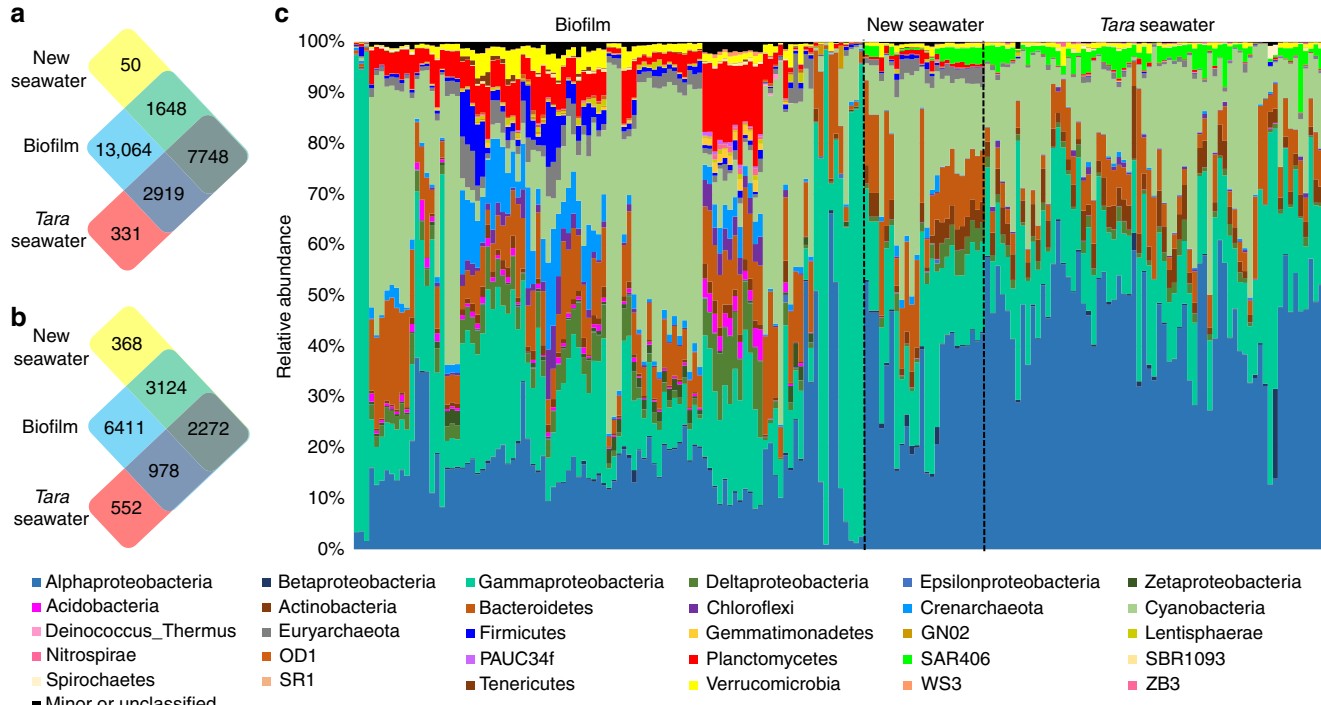

**Fig. 1** Comparative taxonomic analysis of the 101 biofilm and 24 seawater metagenomes sequenced as part of the current study and the 67 previously published *Tara* Oceans metagenomes. **a** Venn diagram showing the distribution of all OTUs. **b** Venn diagram showing the distribution of abundant OTUs (OTUs with more than nine tags in at least one sample). **c** Phylum-level taxonomic structure revealing marked differences in the community profiles between the biofilm and seawater habitats. Relative abundance of the phyla was calculated based on 16S miTag numbers. Abundant phyla (the top 30 phyla in terms of maximum relative abundance) are shown with all other phyla grouped together as "Minor". The order of samples in this figure is the same as that in Supplementary Data 1

16S rRNA gene sequences were extracted from the metagenomes for analysis (subsequently referred to as 16S miTags). Classification of 16S rRNA gene operational taxonomic units (OTU) at 97% similarity resulted in a total of 13,064 OTUs that were exclusively detected in biofilms, 50 OTUs that were unique to the metagenomes of the newly-collected seawater, and 331 OTUs that were unique to the metagenomes of the seawater used in the *Tara* Oceans study (Fig. 1a). When we restricted the analysis to only those OTUs that were abundant (i.e. OTUs with over nine

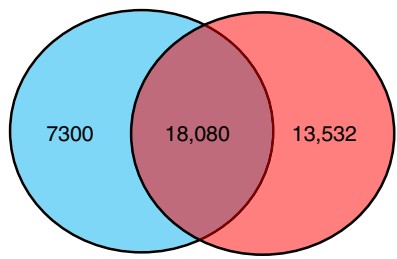

**Fig. 2** Venn diagram showing the distribution of OTUs across the biofilms in the current study (blue) and the *Tara* miTags (red) comprising 16S rRNA sequences from 243 seawater samples

sequences in at least one sample), then 6411 of the biofilm OTUs were niche-specific (Fig. 1b). Classification of OTUs at the phylum level (class level for Proteobacteria) revealed distinct taxonomic patterns between the biofilms ($n = 101$) and the seawater samples ($n = 91$) (Fig. 1c). We also compared biofilm metagenomes with 16S miTags extracted from all reported *Tara* Oceans data (243 metagenomes yielding 31,612 OTUs) which revealed 7300 OTUs undetected in the *Tara* Oceans study (Fig. 2).

Accumulation curves (gamma-diversity analysis) showed that the rate of biofilm OTU detection dropped to 0.2% by the end of sampling (Supplementary Fig. 4), confirming that sufficient sampling had been undertaken to analyze microbial diversity in the marine biofilms. To test the effect of sequencing depth on the discovery of biofilm-specific OTUs, we additionally deeply sequenced two biofilm samples and two seawater samples collected from the same site at the same time (Hong Kong seawater; Dec, 2017), generating more than 110 gigabytes of data per sample. Based on extraction of 20, 40, 80, and 160 million reads, the specificity of the biofilms (the ratio of biofilm-specific OTUs to the total number of OTUs present in the biofilm and seawater samples) increased slightly with sequencing depth, following a linear relationship, whereas seawater samples decreased with sequencing depth (Supplementary Fig. 5),

**Fig. 3** Alpha- and beta-diversity. **a** ACE, Chao1 diversity, and observed OTUs are significantly different between the biofilm-associated (blue) and seawater-derived (red) microbial communities. ***p-value < 0.001 (two-tailed Student's *t*-tests after Shapiro–Wilk test to confirm the normal distribution of these data). In a boxplot, central line represents the median, bounds represent upper and lower quartiles, and whiskers represent maximum and minimum. **b** Jaccard similarity of the microbial communities illustrated by PCoA of the OTU matrix. These analyses were performed after normalizing the different metagenome-derived 16S rRNA gene data to the same library size, i.e. 10,000 sequences per sample

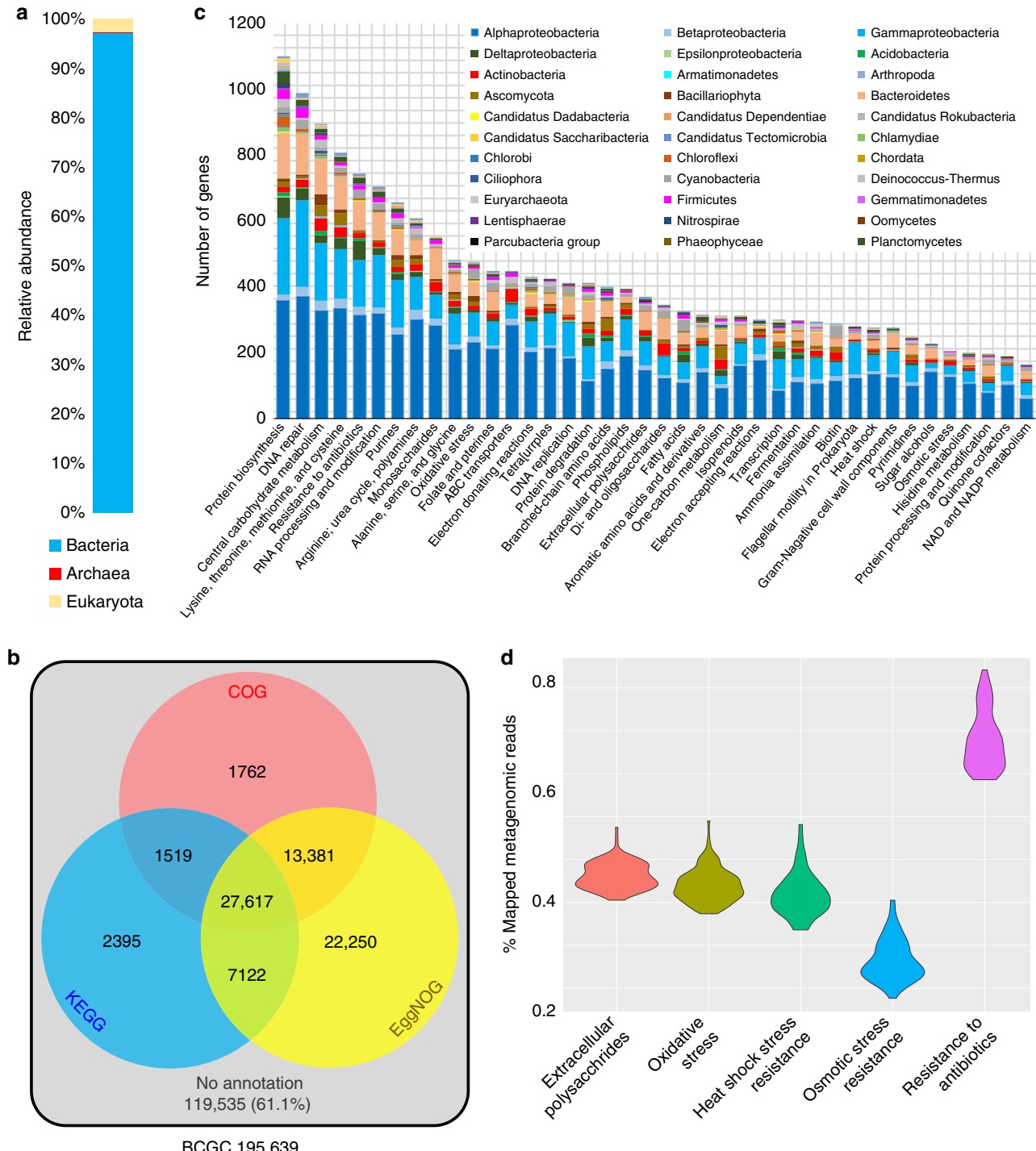

**Fig. 4** Functional structuring of the biofilm core gene catalog (BCGC). **a** Taxonomic breakdown of BCGC genes, 97.3% of which belong to bacteria. **b** BCGC is a nonredundant database comprising genes that are present in more than 99 biofilms but which were not detected in any of the seawater samples. Venn diagram showing the total number of genes in the BCGC and the number of genes annotated by three databases. 61.1% of the BCGC genes could not be annotated. **c** Distribution of the annotated BCGC genes in different SEED categories and their taxonomic affiliation. The 40 most enriched SEED categories and their phylum-level affiliations are shown. **d** Reads mapping to BCGC genes in SEED categories related to extracellular polysaccharide biosynthesis, stress and antibiotic resistance, showing the abundance distribution of these functions across the 101 biofilms

suggesting that additional sequencing would reveal additional biofilm-specific species. Based on analysis of 160 million reads, 33.1% of the biofilm OTUs could not be detected in seawater.

To complement the analysis of 16S miTags, we identified protein-coding marker gene sequences for taxonomic profiling, which confirmed biofilms had different phylum-level composition compared to the seawater microbial communities (Supplementary Fig. 6). Phylum-level composition based on protein-coding marker genes differed from those based on 16S miTags, possibly because the genomic 16S copy number varies substantially across the tree of life[19]. To further examine the contribution of 16S rRNA gene copy number and gene length to

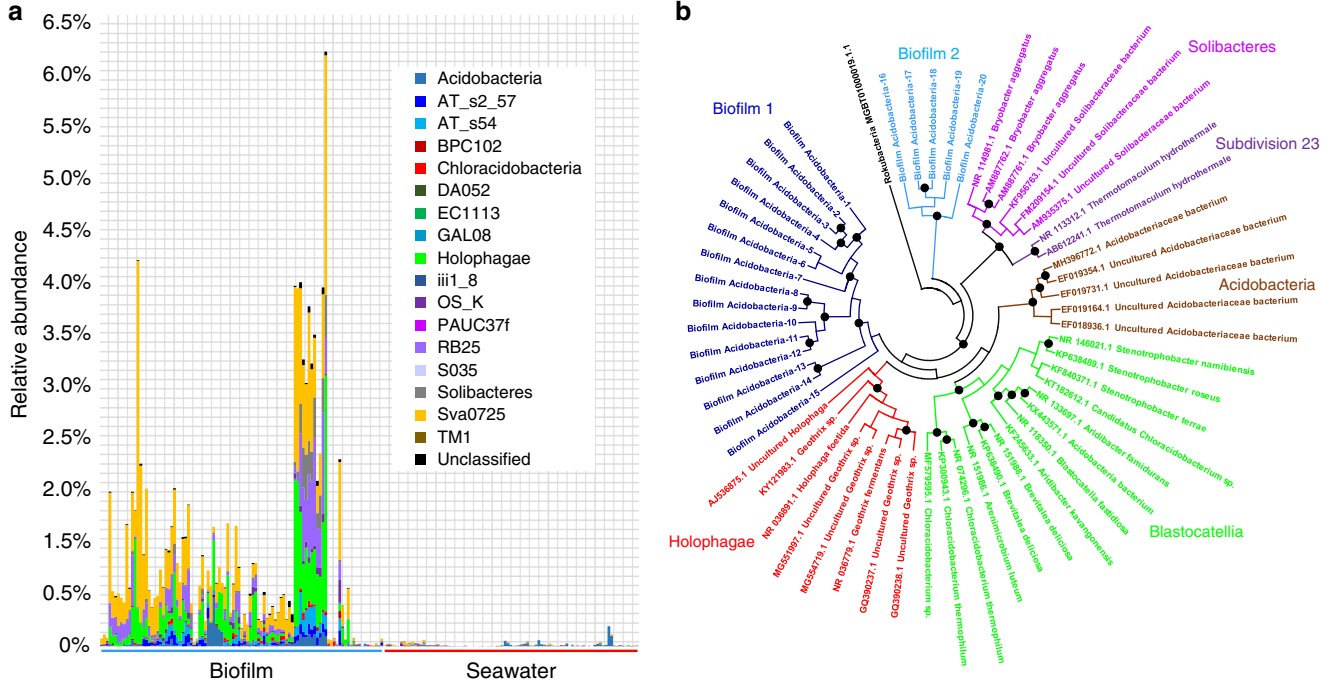

**Fig. 5** Novel Acidobacteria lineages. **a** Distribution of Acidobacteria classes in the biofilms and seawater. Based on 16S miTags, several rarely studied or unclassified groups are enriched in biofilms. **b** Phylogenetic analysis of Acidobacteria 16S rRNA gene sequences (over 1000 bp) extracted from assembled biofilm metagenomes. 16S rRNA gene sequences that are available for known Acidobacteria classes were downloaded from the NCBI database and used as references and Rokubacteria was used to root the tree. Black dots indicate bootstrap values above 60. These analyses provide further support for novel microbial lineages within ocean biofilms

explaining the discrepancy between 16S miTag and protein-coding marker gene data, we normalized the data using CopyRighter, and profiled the microbial community structures again (Supplementary Fig. 7). For certain phyla, the relative abundance derived from protein-coding marker genes was more consistent with that of the normalized 16S miTag data (Supplementary Fig. 8). However, regardless of normalization, results derived from 16S miTags and protein-coding marker genes deviated substantially, consistent with results of a systematic study of 16S rRNA gene copy number correction[20]. While we were unable to resolve the discrepancy amongst approaches within this study, variation in predicting community structure could be attributed to incompleteness of the 16S rRNA gene copy number and protein-coding marker gene databases, which lack information for many uncultured microbes.

Alpha- and beta-diversity analysis based on rarefied 16S miTags (10,000 miTags per sample) from the biofilm and seawater microbiomes revealed a median OTU richness of 2790 and 2155, respectively (Fig. 3a). The median abundance-based coverage estimators (ACE, Chao1, and Shannon diversity were higher for biofilms than for seawater and intra-habitat variation was also greatest in the biofilm samples (Fig. 3a). Principal coordinate analysis (PCoA) using rarefied (Fig. 3b) and total (Supplementary Fig. 9) 16S miTag data both revealed clear separation of the biofilm communities from the planktonic communities, confirming the taxonomic specificity of the biofilms. In addition, PCoA of samples collected from Hong Kong water in 2017 which comprised (i) biofilms developed on artificial panels, (ii) biofilms established on natural rocks, and (iii) seawater microbial communities, revealed larger dissimilarity between biofilms established on artificial panels and seawater than between biofilms occurring on natural rocks and seawater (Supplementary Fig. 10). This analysis suggests that "artificial

communities" may contribute more to the discovery of novel taxa than biofilms on natural substrates.

**Functional core analysis and experimental evidence.** A high level of endemism was evident in biofilm communities at the protein-coding gene level. On average, 57.7% of the open reading frames (ORFs) derived from the 101 biofilm metagenomes had no hits in the OM-RGC catalog or the newly collected seawater gene catalog from the current study (Supplementary Fig. 11). After deprelication, 11,198,462 protein-coding genes remained and formed a biofilm-specific gene catalog (BSGC). Generation of BSGC distribution profiles in each of the 101 biofilm metagenomes revealed a ubiquitous core (genes present in over 99 biofilms), defined as the biofilm core gene catalog (BCGC) comprising 195,639 nonredundant protein-coding genes.

Taxonomic breakdown of the BCGC showed that 97.3% of the genes belonged to bacteria, 2.5% to eukaryotes, and less than 0.2% to archaea (Fig. 4a). Only 38.9% of the BCGC ORFs had hits in the COG, KEGG, and/or SEED databases (Fig. 4b). Of these annotated genes, the 40 most abundant SEED categories included numerous functions involved in biofilm formation, stress response, and environmental adaptation, such as extracellular polysaccharide biosynthesis, DNA repair and resistance to oxidative stress, heat shock, osmatic stress and antibiotics; and these genes were distributed across a variety of phyla (Fig. 4c). Moreover, the abundance distribution of these genes was consistent across the 101 biofilms (Fig. 4d). Clustering of the BCGC protein domains based on hidden Markov models (HMM) classified genes into 2567 Pfam domains, with 91.0% of the BCGC genes having conserved domains including genes conserved in stress resistance and antibiotic resistance, such as the response regulator (PF00072), HSP90-like ATPase (PF02518), AcrB/AcrD/AcrF family genes (PF00873), oxidoreductase (PF01408), and

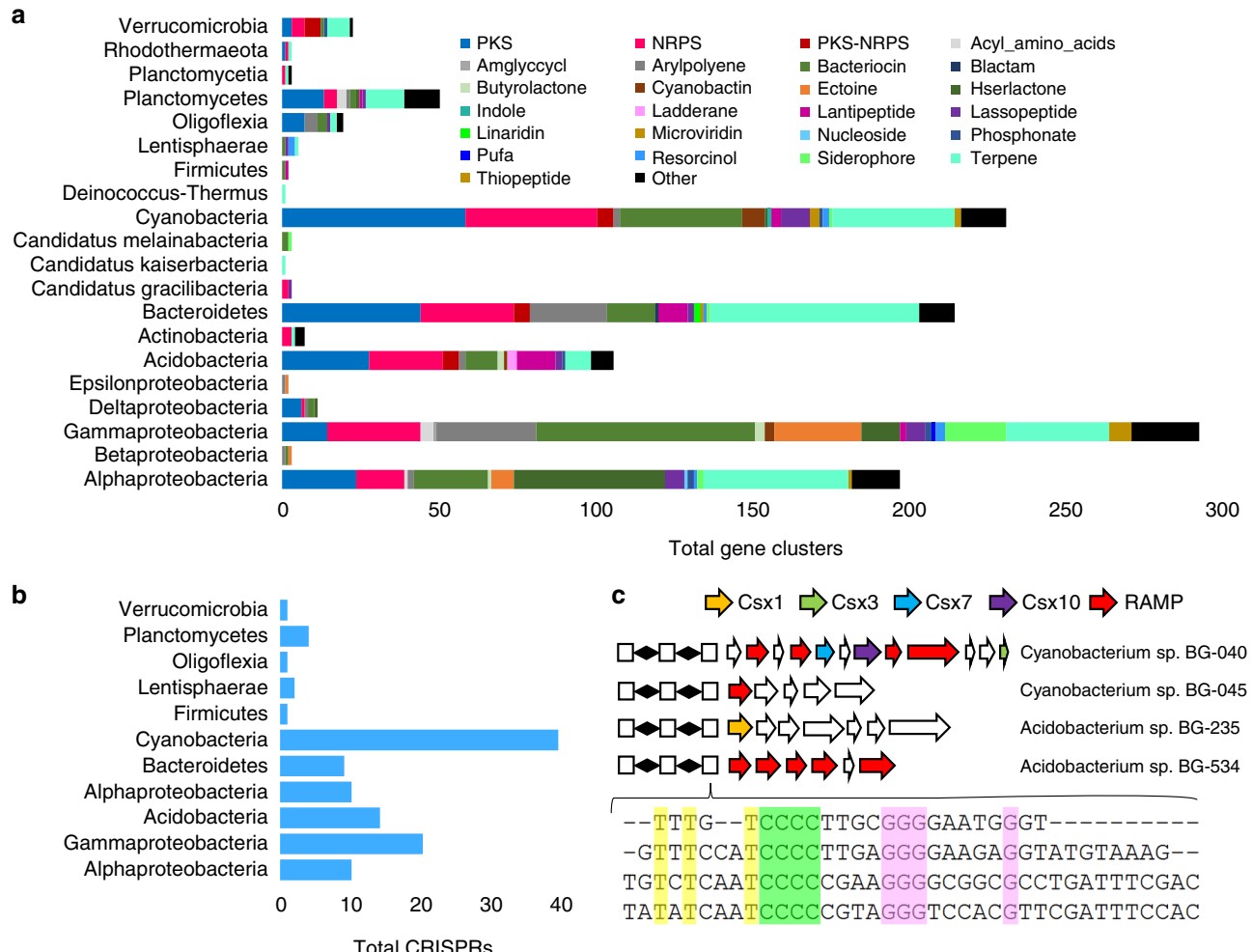

**Fig. 6** Functional potential of the biofilm-forming microbes as revealed by analysis of 479 microbial genomes. **a** Biosynthetic gene clusters found on the 479 genomes recovered from 101 biofilm metagenomes. Putative product types were assigned by antiSMASH. **b** Number of CRISPR arrays identified from biofilm-derived genomes. **c** CRISPR arrays reconstructed from two cyanobacterial genomes and two acidobacterial genomes. White boxes indicate repeats and diamonds indicate spacers. Conserved T, C, and G nucleotides are displayed in yellow, green, and pink, respectively. The putative CRISPR-associated genes are shown with different colors, most of which are RNases. RAMP repeat associated mysterious protein

metallo-beta-lactamase (PF00753) (Supplementary Fig. 12). The AcrB/AcrD/AcrF family of genes encode efflux transporters such as enterobactin that mediate resistance to a number of drugs[21]. Metallo-β-lactamases were first identified in a strain of *Bacillus cereus*[22] and are responsible for multidrug resistance[23]. Given the identification of antibiotic resistance genes, we searched the BCGC against the MEGARes database and found 18 types of antibiotic resistance (Supplementary Fig. 13) arising from 852 genes.

The habitat specificity evident in the biofilm communities suggests that certain taxa and functions with exceptionally low abundance in seawater are enriched during transition from a mobile state to a surface-attached lifestyle. To experimentally test this, laboratory-cultured biofilm communities (Supplementary Fig. 14) and free-living communities derived from the same subtidal seawater, were sequenced. Based on two biological experiments, 918 OTUs were exclusively detected in the biofilm samples (Supplementary Fig. 15a) and of the abundant OTUs (OTUs with over nine sequences in at least one sample), 322 were also unique to biofilms (Supplementary Fig. 15b). Functional comparison revealed that 143 COGs were specific to the biofilms (Supplementary Fig. 16a); 47 of which were uncharacterized genes, and COGs related to signal transduction (e.g., histidine

kinases) and antibiotic resistance (e.g., beta-lactamases) were also identified (Supplementary Fig. 16b), consistent with the finding that genes related to signal transduction become enriched during transition from free-living to surface-associated states.

**Novel microbial lineages and functional potential.** At the class level, classification of Acidobacteria using 16S rRNA gene sequences from assembled biofilm metagenomes revealed several rarely studied or unclassified groups (e.g., AT_s2_57, AT_s54, and BPC102) (Fig. 5a). Phylogenetic analysis including classified Acidobacteria classes (e.g., Solibacteres, Acidobacteria, and Blastocatellia) revealed that the biofilm-derived Acidobacteria are located on two independent branches of the tree, indicating the existence of previously unexplored lineages (Fig. 5b).

The taxonomic and functional specificity of ocean biofilms led us to further explore novel functional potential through genomic analysis. From the 101 biofilm metagenomes, we recovered 479 partial microbial genomes (only contigs over 10 kb were used). These microbes belonged to 25 different microbial phyla (Supplementary Fig. 17), including Alpha-, Beta-, Gamma-, Delta-, and Epsilonproteobacteria, Acidobacteria, six 'Candidatus' phyla, Euryarchaeota, amongst others. Within these contigs, we

identified 1148 biosynthetic gene clusters, distributed across 20 microbial phyla (Fig. 6a). Within the identified gene clusters, 153, 194, and 20 were inferred to synthesize nonribosomal peptides (NRPS), polyketides (PKS), and NRPS-PKS hybrids, respectively. When searching against the MIBiG[24] database, 85.6% of the NRPS and PKS biosynthetic proteins showed <60% similarity. The largest number of gene clusters came from Gammaproteobacteria, followed by Cyanobacteria and Bacteroidetes. While only six genomes belonged to Acidobacteria, 103 biosynthetic gene clusters were predicted from these genomes. One acidobacterial genome encoded long NRPS or PKS gene clusters (Supplementary Fig. 18). On the 8-Mb contig from this bacterium, we identified 14 NRPS or PKS biosynthetic loci that were 846.9 kb in total length, and the longest loci was 127.3 kb. We then sought to identify CRISPR-Cas systems in the microbial genomes, finding 116 CRISPR arrays distributed across 11 bacterial phyla (Fig. 6b). The largest number of CRISPR arrays belonged to Cyanobacteria, followed by Gammaproteobacteria and Acidobacteria. Searching against the CRISPRdb database[25] using CRISPR repeat sequences resulted in only 12 significant hits (e-value < 1e−5). Through mining the CRISPR-Cas loci, we discovered several loci with similar repeats. For example, a CCCC/GGG motif was discovered within CRISPR repeats from two cyanobacterial and two acidobacterial genomes (Fig. 6c). Examining sequences adjacent to these four CRISPR arrays revealed several putative RNases, including Csx1, Csx3, Csx7, Csx10, and repeat associated mysterious proteins (RAMPs) (Fig. 6c), for which BLASTp searches revealed weak similarity (<40%) to known genes in the NCBI-Nr database. For example, the Csx1 protein in Acidobacterium sp. BG-235 possessed two relatively conserved regions with known Csx1 proteins (Supplementary Fig. 19a) while phylogenetic analysis located it in an independent branch of the tree (Supplementary Fig. 19b).

## Discussion

The OM-RGC catalog generated by *Tara* Oceans revolutionized our understanding of oceanic microbial diversity and functional potential and provided a platform to resolve the ecological roles of the microbial community. 16S miTags derived from the *Tara* Oceans metagenomic datasets mapped to a total of 35,650 OTUs, with the rate of new gene detection only 0.01% by the end of seawater sampling[4]. However, by sampling diverse biofilm surfaces assembled from seawater across broad spatial and temporal scales and performing comparisons with the OM-RGC, we revealed a large reservoir of novel microbial OTUs and functional genes not previously considered part of the global ocean microbiome. Overall, metagenomic analyses of these biofilms increased the known microbial diversity of the global oceans by more than 20%. Of the biofilm ORFs, 57.7% had no hits in the OM-RGC catalog or the newly collected seawater gene catalog, providing further support for underestimated diversity in the ocean microbiome.

Analysis of the taxonomic and functional core of the biofilm-forming microbiome revealed the existence of persistent microbial groups and associated functions. The findings of beta-diversity analyses are noteworthy given that biofilms were collected across such broad spatial and temporal scales and included diverse substrate surfaces. The BCGC contained endemic genes with conserved domains, suggesting that gene sequences may have evolved along divergent paths without affecting the functional domains. Stress response and antibiotic resistance genes appear to be important components of the biofilm functional core. Stress response contributes to bacterial survival in natural environments and can trigger biofilm formation[26]. The function of antibiotic resistance genes in natural microbial communities is often uncertain, as their abundance is not solely determined by the presence of these antibiotics[27]. However, antibiotics have been shown to act as signaling molecules within natural microbial communities[28], and thus antibiotic resistance genes may play roles outside of the 'war' metaphor, consistent with recent findings in soil microbial communities[29]. Our laboratory experiment also highlighted the role of specific functions in determining niche differentiation between surface-associated and planktonic communities. We surmise that the biofilm core gene set underpins environmental adaptation and microbe–microbe interaction.

The biofilm microbiome also serves as a resource for the discovery of novel microbial lineages and functional genes. Previous analyses based on publicly available acidobacterial genomes identified few biosynthetic clusters[30–32], whereas a recent study showed that Acidobacteria from soil express unusually large PKS and NRPS biosynthetic genes[33]. Here we show that Acidobacteria in biofilms formed in the marine environment also contain numerous biosynthetic gene clusters. In addition, we identified several PKS biosynthetic genes in Oligoflexia which had not previously been linked to secondary metabolite production. In recent years, a wealth of sequence information has accumulated for CRISPR-Cas, facilitating discovery of new systems and novel functions[34]. Here we identified a number of CRISPR arrays in the biofilm microbial genomes, consistent with an expectation for enhanced interactions between viruses and bacteria within biofilms[35,36]. The finding of CRISPR loci with conserved repeats in cyanobacterial and acidobacterial genomes and the presence of adjacent putative RNases suggests the existence of conserved but functionally unknown CRISPR-Cas systems in marine biofilms.

While surface-associated microorganisms must be seeded from the seawater microbiome, many microbial species in seawater are simply too scarce to be captured in global sequencing efforts and can only be detected when their relative abundance increases during biofilm formation. Here we demonstrate the important niche differentiation that occurs between planktonic and surface-associated communities and show that marine biofilms represent a novel species bank of hidden microbial diversity and functional potential in the ocean.

## Methods

**Sampling**. Biofilms were developed at a depth of 1–2 m at eight locations spanning Sapelo Island in the South Atlantic, Red Sea, Hong Kong Seawater, Yung Shu O Bay, East China Sea, South China Sea 1, South China Sea 2, and South China Sea 3 (Supplementary Fig. 1). Biofilms developed in Hong Kong seawater were collected annually between 2013 and 2017. Artificial panels and polystyrene Petri dishes were used to establish the biofilms: 9 × 1.2 cm sterile polystyrene Petri dishes (Supplementary Fig. 2a), 11 × 11 cm zinc panels used for preventing marine biofouling (Supplementary Fig. 2b), and 5 × 5 cm panels of aluminum, poly(ether-ether-ketone), polytetrafluoroethylene, poly(vinyl chloride), stainless steel, and titanium (Supplementary Fig. 2c) held in place by a customized device (Supplementary Fig. 2d). Most biofilms were fully established on Petri dishes after 12 d, and 30 d were required for visible biofilm formation on the metal surfaces. Biofilm development was limited to 3 d in the South China Sea for logistical reasons. For biofilms collected in Jun, Jul, Aug, and Sep, the panels were covered with mesh to prevent settlement of macroorganisms. Biofilms were also removed from rocks present in the subtidal zone (Supplementary Fig. 2e). Collected biofilms were immediately transferred to the laboratory, and the biomass was removed using sterile cotton tips and stored at −80 °C in 5 ml of DNA storage buffer (500 mM NaCl, 50 mM Tris-HCl, 40 mM EDTA and 50 mM glucose). Samples were observed under a Zeiss LSM 510 confocal laser scanning microscope (Carl Zeiss, Jena, Germany) as previously described[37]. The observation revealed thick films with multiple cell layers, thus confirming microbial attachment (Supplementary Fig. 3). Prior to the observation, microorganisms were labeled with the protein dye fluorescein isothiocyanate (FITC) (Sigma, Poole, UK). Images were generated using a 63 × 1.4 objective, and signals were recorded in the green channel (excitation 488 nm, emission 522/32 nm). Adjacent seawater samples were collected at the same time as biofilm retrieval and successively filtered through 0.22-μm and 0.1-μm polycarbonate membrane filters (Millipore, Massachusetts, USA). The filters were stored in 5 ml of DNA storage buffer at −80 °C. In total, 101 biofilm and 24 seawater samples were collected. Two additional seawater samples were collected for a laboratory experiment on biofilm formation (see below for details).

**DNA extraction**. Before DNA extraction, 10 mg/ml lysozyme was added to the samples and incubated at 37 °C for 30 min. To test the efficiency of DNA extraction, a subset of biofilms was also extracted with the TIANamp Genomic DNA Kit (Tiangen Biotech, Beijing, China), the AllPrep DNA/RNA Mini Kit (Qiagen, Hilden, Germany), and the DNeasy Blood & Tissue Kit (Qiagen, Hilden, Germany), following the Manufacturer's protocols; in total, six biofilms were collected from two Hong Kong seawater sites, with three biofilms collected per site for each of the three extraction methods. The TIANamp Genomic DNA Kit was used to extract DNA from all other biofilm and seawater samples.

**Metagenomic analyses**. Most samples were sequenced at the Novogene Bioinformatics Institute (Novogene, Beijing, China) and a subset was sequenced at the Beijing Genomics Institute (BGI, Beijing, China). After construction of 350 bp insert libraries, metagenomic samples (including both biofilms and seawater) were pair-end sequenced on the HiSeq X Ten System at Novogene and the HiSeq 2500 System at BGI. The read length of the data generated on both platforms was 150 bp. Metagenomes were generated for 101 biofilm and 24 adjacent seawater metagenomes, with $115 \pm 39$ million reads per biofilm metagenome and $130 \pm 33$ million reads per seawater metagenome. Quality control was performed on a local server using the software NGS QC Toolkit (version 2.0)[38]. Reads containing adaptors, low quality reads (assigned by a quality score <20 for >30% of the read length) or unpaired high-quality reads were removed. Following quality control, short reads from the biofilm and seawater metagenomes were assembled into contigs using the software MEGAHIT (version 1.0.2)[39] with kmer values from 21 to 121. The OM-RGC and 67 metagenomes of surface seawater in the *Tara* Oceans study were downloaded for comparison, with $302 \pm 114$ million reads per metagenome. Detailed information about samples and corresponding sequencing data is shown in Supplementary Data 1. Information on assembled metagenomic contigs is shown in Supplementary Data 2.

**Taxonomic profiling**. 16S miTags were extracted from the unassembled metagenomic data using Parallel-META3[40], which relies on HMMER (version 3.1)[41] to predict the 16S miTags from both the forward and reverse sequences. Bowtie2[42] was used to map the 16S miTags to a database that integrates GreenGenes[43] with RDP[44] and SILVA[45] (assigned by BLASTn with e-value < 1e−30 and similarity >97%). OTUs were picked at 97% similarity or above. To reduce the false discovery rate due to sequencing errors, only OTUs identified in both forward and reverse paired reads in a sample were retained (thus all singletons were removed). Considering the primary objective of this analysis was to estimate the total number of biofilm-specific species and considerably more data exists for seawater metagenomes than for the biofilm metagenomes, OTU accumulative analysis of the 101 biofilms was performed using the 'vegan' package in R with 100 permutations. The full OTU list generated by the 16S miTags and used for Venn diagram comparison is given in Supplementary Data 3. The full list of phyla in biofilms and seawater based on 16S miTag analysis is given in Supplementary Data 4. To assess the effect of 16S rRNA gene copy number and gene length on taxonomic classification, results derived from the 16S miTags were normalized using the software CopyRighter[46], which performs lineage-specific gene length and gene copy number correction. The databases genlength_img40_gg201210 and ssu_img40_gg201210.txt were used. The full list of phyla in biofilms and seawater based on normalized 16S miTags is given in Supplementary Data 5. Metagenomic reads were also mapped to the mOTU.v1.padded database consisting of 40 marker genes for taxonomic profiling as previously described[47]. Phylum-level profiles were generated for the 101 biofilm and 91 seawater samples. The list of phyla in all biofilm and seawater microbial communities based on these marker genes is given in Supplementary Data 6.

**Sample accumulation curve and sequencing depth analysis**. The accumulation curve of detected OTUs in the 101 biofilms was calculated using the 'vegan' package in R with 100 permutations. The OTU table based on the 16S miTags was used as the query. To determine how specificity was influenced by sequencing depth, two biofilms were developed on a Petri dish for 12 days in Hong Kong seawater in Dec 2017 and two seawater samples were collected adjacent to the sites of biofilm development. Metagenomic sequencing was performed on the HiSeq X Ten System in Novogene (Beijing, China) to generate 110 Gb data. The seqtk program (https://github.com/lh3/seqtk) was used to randomly extract 20 M, 40 M, 80 M, and 160 M reads from the metagenomes. The extracted reads were used for 16S miTags identification and OTU classification. The biofilm and seawater specificity was calculated based on the formulas:

$$\text{Biofilm specificity} = \frac{\text{Number of biofilm-specific OTUs}}{\text{Number of OTUs in biofilm and/or water}}$$

$$\text{Water specificity} = \frac{\text{Number of water-specific OTUs}}{\text{Number of OTUs in biofilm and/or water}}$$

**Alpha- and beta-diversity analyses**. Alpha- and beta-diversity analyses were performed on the 101 biofilm and 91 seawater metagenomes. To normalize data

size, 10,000 16S miTags were extracted from each sample (only forward sequencing files were used). OTUs were picked at 97% similarity or above. ACE, Chao1, observed OTUs, and Shannon diversity were calculated using scripts (alpha_diversity.py and default parameters) implemented in QIIME[48]. Jaccard distances, which are based on the presence/absence of taxa, were generated using the OTU table for pairwise comparison of the biofilm and seawater communities. Jaccard distances were revealed by PCoA using the software PAST (version 2.0)[49]. In addition, PCoA was performed using the total 16S miTags. The OTU tables constructed using rarefied (10,000 per sample) 16S miTags and total 16S miTags are presented in Source Supplementary Data 7 and 8, respectively.

**Identification and annotation of the biofilm functional core gene set**. ORFs were predicted from the contigs using MetaGeneMark (version 2.8) (predict genes on both strands; gene overlaps are allowed; probability of initiation and termination in non-coding state 0.5). Protein sequences (minimum length 100) derived from the 24 seawater metagenomes were combined and subjected to CD-HIT[50] (>95% sequence identity for >60% of the length of the shorter sequences) to generate a NSGC comprising 2,541,441 genes. Protein sequences (minimum length 100) derived from the biofilm metagenomes were BLASTp (e-value < 1e−7; >60% sequence identity for >60% of the length of the shorter sequences) searched against OM-RGC[4] and the new seawater gene reference catalog generated in this study. Proteins without BLASTp hits were then searched against the OM-RGC and the new seawater gene reference catalog using USEARCH (global alignments)[51] to confirm BLASTp results. Protein sequences without BLASTp or USEARCH hits were subjected to CD-HIT (>95% sequence identity for >60% of the length of the shorter sequences) to generate a nonredundant catalog called BSGC. The BSGC genes that were present in more than 99 of the 101 biofilms were defined as the BCGC. For functional annotation, BCGC proteins were DIAMOND[52] BLASTp (e-value < 1e−7; >60% sequence identity for >60% of the length of the shorter sequences) searched against the COG[53], KEGG (2016 version)[54], and eggNOG databases[55]. eggNOG individual gene annotations and functional category profiles of the BCGC were generated by linking the eggNOG annotations with the SEED subsystems database[56] using mapping files documented in MOCAT[57]. Moreover, the BCGC protein sequences were searched against the Pfam database[58] using HMMER hmmscan (e-value < 1e−7) to identify conserved functional domains. Finally, the same protein sequences were BLASTp searched against the MEGARes database[59] (e-value < 1e−7; >60% sequence identity for >60% of the length of the shorter sequences) to identify antibiotic resistance genes. Data from all BCGC gene annotations (COGs, KEGG, SEED, and Pfam) are presented in Supplementary Data 9. The full list of antibiotic resistance genes is provided in Supplementary Data 10.

**Laboratory experiment on biofilm formation**. Hong Kong seawater samples were collected in duplicate, 10 m apart in Jan 2018. Two-hundred milliliters of each sample were filtered through 0.1-μm polycarbonate membrane filters. Microbes were washed from the membranes using 20 ml of 0.1-μm filtered and autoclaved seawater and each sample was divided into two parts for parallel experiments. In treatment 1 (biofilm-forming culture), 10 ml of each aliquot was placed in the modified M9 minimum medium (prepared using 0.1-μm filtered and autoclaved seawater) and incubated at 22 °C for 24 h in the wells of a six-well plate (BD Biosciences, California, USA). Films that formed on the bottom surface of the plate were removed for DNA extraction and metagenomic sequencing. In treatment 2 (free-living culture), the remaining 10 ml of each aliquot was suspended for 24 h in 50 ml falcon tubes in an incubator shaker (22 °C, 100 rpm). Microorganisms in the falcon tubes were collected by centrifugation at $4000 \times g$ for 10 min for DNA extraction and metagenomic sequencing. Taxonomic analysis was performed based on 16S miTags, as described above. Functional analysis was performed using searches against the COG database. The OTUs and protein-coding genes are given in Supplementary Data 11 and 12, respectively.

**Mining novel lineages**. 16S rRNA genes (over 1000 bp) were extracted from contigs based on HMM using the software Meta_RNA[60]. The 16S rRNA gene sequences of representative members in all classes of Acidobacteria were downloaded from GenBank. The 16S rRNA gene sequence retrieved from a genome of Rokubacteria, which are close relatives of Acidobacteria in the tree of life[61], was used to root the tree. The maximum-likelihood phylogenetic tree was constructed in MEGA (version 7.0)[62] using the Tamura-Nei substitution model and bootstrapping was performed with 1000 repetitions.

**Genome binning and analyses**. Genome binning was performed with contigs longer than 10 kb. A total of 105,905 contigs were clustered into genome bins using a hybrid binning approach. Initially, reads from a single metagenome were mapped to the corresponding assembly using Bowtie2 (fastq as query input files) to generate a coverage profile, which was used to inform the binning software MaxBin2[63] (minimum probability for EM algorithm 0.8; 107 marker genes) to separate the contigs. Subsequently, these contig groups were individually subjected to a second binning process using MetaBAT[64]. The resulting output genome bins were examined for completeness and potential contamination using CheckM[65] (lineage_wf). FetchMG[47] (version 1.0) was used to identify single-copy genes from the

genome bins to check for potential contamination and analyze the taxonomic affiliation using default bit scores. Closely related genome bins were compared based on average nucleotide identity (ANI)[66]. Genomes with an ANI value higher than 0.99 were considered duplicates and the most complete genome bin was retained. Genome bins were processed using antiSMASH (version 4.0)[67] (default parameters) for biosynthetic gene cluster identification. Proteins involved in NRPS or PKS biosynthesis were searched against the MiBIG repository[24] using BLASTP (e-value < 1e−7; >60% sequence identity for >60% of the length of the shorter sequences) for similarity assessment. CRISPR arrays were identified using the CRISPRs mining tool MinCED[68] (minimum number of repeats a CRISPR 3; minimum length of the CRISPR repeats 23; maximum length of the CRISPR repeats 47; minimum length of the CRISPR spacers 26; maximum length of the CRISPR spacers 300) and selectively checked using the online tool CRISPRFinder[69] (http://crispr.i2bc.paris-saclay.fr/Server/) using default parameters. Proteins adjacent to the CRISPR arrays were searched against the online NCBI-Nr database for annotation. The Csx1 protein sequence and its top hits in the NCBI-Nr database were included for alignment and phylogenetic analysis using MEGA (version 7.0)[62]. A maximum-likelihood tree was constructed using the Jones-Taylor-Thornton substitution model, bootstrapping with 500 repetitions. Genome information, including taxonomic affiliation, genome completeness, and potential contamination is shown in Supplementary Data 13. Total biosynthetic gene clusters and CRISPR arrays are given in Supplementary Data 14 and 15, respectively.

**Reporting summary**. Further information on experimental design is available in the Nature Research Reporting Summary linked to this article.

## Data availability

All metagenomic datasets have been deposited in the NCBI database under Bio-Project accession no. PRJNA438384 (BioSample no. SAMN08714533 for the 101 biofilm metagenomes; BioSample no. SAMN0871453 for the 24 adjacent seawater metagenomes; BioSample no. SAMN08714627 for the two biofilm and two seawater metagenomes that have been sequenced for more than 110 Gb data; and BioSample no. SAMN08714628 for the metagenomes of the laboratory cultured biofilm and free-living microbes). Gene catalogs, including the biofilm-specific genes and core genes, as well as the seawater gene sequences are uploaded to figshare (BSGC, https://figshare.com/s/7e193dceb329c02cb166, https://doi.org/10.6084/m9.figshare.6743327; BCGC, https://figshare.com/s/53f27d05d443c6c947e5, https://doi.org/10.6084/m9.figshare.6743324; NSGC, https://figshare.com/s/7815ea074e5c2d7ef1c5, https://doi.org/10.6084/m9.figshare.6743459). The 479 microbial genomes are uploaded to figshare (https://figshare.com/s/2994fdafe79112b99907, https://doi.org/10.6084/m9.figshare.7082684). Supplementary Figs 1–19 and Supplementary Data 1–15 are present in the supplementary materials.

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

## Acknowledgements

We thank Mr. Huihui Zhu, Ms. Yujia Nie, and Ms. Xin Gong from Novogene for DNA sequencing and technical support. We thank Mr. Bo Yang for technical support in construction of Linux platform. We thank Ms. Qian Ding, Dr. Zhiwu Sun, Dr. Tim Wong, and Dr. Zhaoming Gao for sample collection. The authors are also grateful for English editing by Ms. Alice Cheung. We also thank the fundings provided by the China Ocean Mineral Resources Research and Development Association (COMRRDA17SC01) and the Strategic Priority Research Program of Chinese Academy of Sciences (XDB06010102) to P.Y.Q.

## Author contributions

W.Z. and P.Y.Q. designed the project; W.Z. and W.D. performed major metagenome and 16S data analyses; S.B. and V.B.B. provided Linux platform and are involved in data analyses; Y.L., H.F., and H.L. proposed cruises and collected samples; W.D., C.T., R.W., B.P., H.C., and P.L. collected samples and extracted DNA; W.Z., Y.X.L., N.S.W., and P.Y.Q. wrote the paper; P.Y.Q., J.S., and Y.X.L. provided funding support for DNA sequencing and human resources; all the authors discussed and revised the paper.
