## [Peer Review File · Nature Communications]

Reviewers' comments:

Reviewer #1 (Remarks to the Author):

Review on Zhang et al., 2018

Title: A bank of undetected ocean microbial diversity and functional potential

Authors: Weipeng Zhang, Wei Ding, Yong-Xin Li, Chunkit Tam, Salim Bougouffa, Ruojun Wang, Bite Pei, Hoyin Chiang, Pokman Leung, Yanhong Lu, Jin Sun, He Fu, Vladimir B Bajic, Hongbin Liu, Nicole S Webster, Pei-Yuan Qian

Aim of Zhang et al. was to increase knowledge on the microbial diversity in the Ocean, on taxonomical and functional level. The approach of the authors was to enrich not yet detected members of the seawater communities by stimulating their growth on artificial surfaces in the Ocean (like PVC etc.). Eventually, the authors investigated the microbial composition by generating and analyzing 2.3 terabases sequenced datasets (metagenomes but also more specific 16S rRNA gene analyses) and in principal compared it to the Tara Oceans metagenomes. Alpha- and beta-diversity was determined on the biofilm and seawater metagenomes; moreover, the biofilm functional genes were identified and binned.

As result, more than 7,300 formerly unknown 16S rRNA gene OTUs were detected in the biofilms. Novel lineages include, for instance, the Acidobacteria. They must have derived from seawater and probably belonged to its rare organism fraction which often remains undetected by currently established methods. Zhang et al. expected that the known microbial diversity increased by more than 20% using their biofilm approach. They also identified a functional core across the biofilms likely play a role in stress responses and microbe-microbe interactions.

Methods used to enrich members of the rare communities as well as following diversity analyses are appropriate. This also means that the statement concerning the known microbial diversity, which increased by more than 20%, is in principle supported.

However, other conclusions remain unclear to me. For instance, what is meant by "high level of endemism was also evident through analyzing biosynthetic gene clusters and CRISPR-Cas systems in 479 genomes extracted from the biofilm metagenomes" (lines 24-26)? The authors used several artificial substrates which are no natural part of the Ocean to enrich unusual microorganisms. Do they expect endemic organisms on these artificial substrates or just substrates in general? These questions are getting more and more important and the authors should include the potential role of a "plastisphere" into their discussion. To go further, could artificial substrates even impact evolution and by this generate "artificial communities"? Taken together, a complex and new research field the authors are touching and the whole manuscript should be carefully revised concerning natural and artificial substrates. Both habitats have been investigated in the study, a more detailed comparison should be provided here.

Minor comments:

Line 71: Please mention explicitly that 16S rRNA genes are meant

Reviewer #2 (Remarks to the Author):

The manuscript NCOMMS-18-28752 submitted by Zhang et al, A bank of undetected ocean microbial diversity and functional potentia, presents an in-depth look at a dataset that greatly expands our view of microbial marine life. The analyses appear to be well executed and appropriate to the description of the data. I see no obvious flaws in the work, and I consider it a significant addition to our understanding of the ocean.

There is not enough detail given in the methods. In general, please provide ALL runtime parameters for software used, even if it's in the supplement. Two specific cases stood out to me:

In the Methods: Metagenome Analysis, you state: “[L]ow quality (quality score below 20) reads . . . were removed.” Quality scores are usually given per base not per read. This could be interpreted as an average score or a minimum score, please clarify. (Based on the software defaults, I’d guess it should be: quality below 20 for more than 30% of the read.)

In the Methods: Identification and annotation of the biofilm functional core gene set, cd-hit appears to be used twice. The first time, a 95% identity cutoff is mentioned, but no parameters are given for the second. Please provide all parameters used, not just the identity. There should have been some match length restrictions applied.

My initial major complaint with the paper paritally melted away when I re-read it just now. It was not clear to me at first if the paper claimed that it was the first comparison of metagenomes between planktonic and attached marine samples or if it is merely the first study to do it at scale. On closer inspection the middle paragraph of the introduction does cite previous work, but the final paragraph (taken on its own) may overstate things a bit. Clarifying the language would help readers understand the impact of the work.

It think it is important to get it right in the introduction in order to properly put into perspective the later claims of the potential trove of data in these metagenomes.

Two typos stood out:

CheckM is misspelled as ChechM.

Citations 5 and 11 look like the same paper. Is that a dupliate or missing citation?

With the addition of all software parameters and clarification of the impact, I would recommend this paper for publication.

-J M Eppley

Reviewer #3 (Remarks to the Author):

Zhang and collaborators present a collection of metagenomic and 16S amplicon datasets from surface-attached and free-living communities in geographically distributed marine locations. The data collection reported here is a valuable resource for the scientific community. However, I have major reservations on the accompanying manuscript and analyses, as detailed next. Note that the list below is intended to highlight the clearest examples of each issue, but is not meant to be comprehensive.

Major Concerns

1. In my opinion, the introduction of this manuscript misrepresents the current state of the art and fails to justify the present study.

1a. As I state above, I believe the data presented in this study are a valuable contribution to the scientific community, but I fail to recognize this value out of the introduction. The aims of the study are presented in L58-62, and their justification in L55-57. However, the ecologic “principle” cited is actually a highly contested (if often cited) conjecture, and no other justification is given.

1b. Additionally, the introduction of the current manuscript would give the (erroneous) impression that this is the first report of this kind, and the first attempt at answering the posited aims. I understand the need for brevity, but failing to mention at the very least the (very similar) study by Bryant et al (*mSystems* 1(3), May 2016) is a major omission of this section. In fact, that manuscript and references within (refs. 27-31 in Bryant et al 2016) clearly demonstrate that, indeed, the populations that form biofilms on plastics are distinct from those in the water column.

2. The inconsistency between results from taxonomic assignments of metagenome-derived 16S reads and 16S amplicons are understated.

2a. The authors recognize the differences between the two methods as presented in Suppl. Fig. 6, in comparison to Fig. 1c (L94-97), but these are simply discarded as an artifact of 16S copy number. No doubt this plays an important role in the difference, but its relative effect is never tested. Methods to account for copy-number variation exist (e.g., see Guo et al, *AEM* 82(1), Oct 2015), and other biases that could be affecting the analyses in this manuscript could play a role too.

3. The inconsistency between results from 16S amplicons and metagenome-derived 16S reads is not properly described or discussed.

3a. In L108-111, the authors indicate that the differences between taxonomic assignments from biofilm and water column communities are consistent across methods. This is not evident from Suppl. Fig. 8c v Suppl. Fig. 8d. In fact, the results indicate strong biases that are never discussed. Notably, the identification of biofilm-specific taxa is one of the main aims of this study, and yet the authors fail to identify the clear differences between the distributions presented in Suppl. Fig. 8a vs Suppl. Fig. 8b. 34% of the metagenome-derived 16S OTUs detected in biofilms are also detected in the water column, but this fraction is only 3% in the 16S amplicons. This indicates a strong bias in

one or both methods that must be addressed. All analyses derived from these OTU assignments are invalidated by the possibility of such a large bias.

3b. Another clear example of these biases resides on the richness estimations. For example, the authors report a total of 25,379 OTUs detected in all 101 biofilm samples, and infer (based on rarefaction analysis) that this number is very close to saturation (L83-85). However, this figure is nearly matched by a single 16S amplicon library (Suppl. Fig. 9), and the total number of OTUs detected in only 9 amplicon libraries from biofilms was 58,615 (Suppl. Fig. 8a). In other words, less than 10% of the samples already show more than twice that number of OTUs, indicating that the sampling is actually very far from saturation. Using only the corresponding 9 metagenomic samples, the authors find only 8,038 OTUs, less than 15% of those observed in the amplicons. Similarly, in L116-117 the authors indicate that the median richness in biofilms was 2,790, about 10 times less than the values shown in Suppl. Fig. 9. These results clearly contradict the statement that “microbial diversity was adequately captured” and further underscore the issues raised in point #2.

4. The assumptions of several key analyses are violated or left untested.

4a. The analysis in Suppl. Fig. 4 was performed on number of samples, not on sample size, and is therefore an indication for gamma diversity, not alpha diversity, but this is not clearly indicated in the text. More importantly, this analysis was performed on classified OTUs, and is therefore biased by the incompleteness of the databases. Although the authors don't report the degree of novelty of the datasets presented (only the differences with respect to the TARA collection), presumably these datasets are not fully captured by existing sequences in the tested databases. This violates the assumptions behind collector curves.

4b. L325-354: “... only OTUs identified in both forward and reverse paired reads in a sample were retained (thus all singletons were removed)”. Does this mean that each pair was counted twice? If yes, several statistical analyses that assume independence of observations are compromised. If not, and the authors meant to indicate that, in addition, singletons were removed, how were the Chao1 estimators of richness calculated? Note that Chao1 uses the number of OTUs seen exactly once (singletons).

4c. L111-113 identifies 500,000 as the number of reads per sample necessary to saturate sequencing, and yet Fig. 3 presents analyses performed on rarefied datasets at only 10,000 sequences per sample. In fact, rarefying samples is not a robust normalization method and should be avoided altogether (e.g., see McMurdie and Holmes, PLoS CB 10(4), 2014). Also, see minor issue #5.

4d. Suppl. Fig. 12 shows that the most abundant category of resistance (after “Unclassified”) was “RequiresSNPConfirmation_1”. No details are provided in the methods or the legend, so it would appear this simply corresponds to genes that may confer resistance with specific SNPs. Was this further examined? These genes may correspond to housekeeping genes, potentially inflating the statistics presented by the authors on antibiotic resistance.

Minor Issues

1. The manuscript creates a good deal of unnecessary confusion by avoiding explicit labeling of 16S amplicons vs miTags. A reader can find all the information with some effort, but clearly stating the base data of each analysis could simplify the reading.

2. The first mention of miTags (L79) is not accompanied by an explanation of what it is.

3. L71: “...resulted in a total of 13,064 unique OTUs in the 101 biofilms...”. This phrasing can introduce some confusion, since many readers could interpret “unique OTUs” as total OTUs after dereplication, as opposed to the (intended) meaning of “OTUs unique to biofilms”.

4. L88-89: “... excluding any impact from DNA extraction bias on community structure of the biofilms”. Not really. In fact, Suppl. Fig. 5 clearly shows that the biases introduced by different methods could be sample-dependent. This means that a much larger number of samples would be necessary to exclude any impact. In addition, the same figure shows that the DN2 has a larger dissimilarity to TIAN2 and AIP2 than these exhibit to sample 1 (with any extraction method). Since there is no ground truth dataset, and the variability introduced by the methods doesn’t appear to be systematic (at least based on those two samples), there is an important probability that the TIANamp method (used in the rest of the study) has a strong impact on (observed) community structure.

5. The authors indicate in L11-113 that saturation is reached at 500,000 reads per sample, but the horizontal axis of Suppl. Fig. 9 only extends to 410,000.

6. L244-245: “We surmise that the biofilm core gene set underpins environmental adaptation and microbe-microbe interaction”. It would be valuable to further test other indicators of interaction such as toxins/antitoxins or Quorum-sensing signals.

7. L88-89 and Suppl. Fig. 5: should be “observed” community structure and taxonomic composition. Neither actual community structure nor the actual taxonomic composition are affected by DNA extraction.

8. L389: USEARCH.

9. L456: “... the Software rRNA prediction”, presumably refers to Meta_RNA.

10. L460-461: “using the maximum likelihood as the substitution model”. Please review the methods carefully, this is a major error indicating that this section was not written carefully, and could give the reader the impression that it wasn’t performed very carefully either.

Signed: Luis M. Rodriguez-R.

Reviewer #1 (Remarks to the Author):

Aim of Zhang et al. was to increase knowledge on the microbial diversity in the Ocean, on taxonomical and functional level. The approach of the authors was to enrich not yet detected members of the seawater communities by stimulating their growth on artificial surfaces in the Ocean (like PVC etc.). Eventually, the authors investigated the microbial composition by generating and analyzing 2.3 terabases sequenced datasets (metagenomes but also more specific 16S rRNA gene analyses) and in principal compared it to the Tara Oceans metagenomes. Alpha- and beta-diversity was determined on the biofilm and seawater metagenomes; moreover, the biofilm functional genes were identified and binned.

As result, more than 7,300 formerly unknown 16S rRNA gene OTUs were detected in the biofilms. Novel lineages include, for instance, the Acidobacteria. They must have derived from seawater and probably belonged to its rare organism fraction which often remains undetected by currently established methods. Zhang et al. expected that the known microbial diversity increased by more than 20% using their biofilm approach. They also identified a functional core across the biofilms likely play a role in stress responses and microbe-microbe interactions.

Methods used to enrich members of the rare communities as well as following diversity analyses are appropriate. This also means that the statement concerning the known microbial diversity, which increased by more than 20%, is in principle supported.

However, other conclusions remain unclear to me. For instance, what is meant by “high level of endemism was also evident through analyzing biosynthetic gene clusters and CRISPR-Cas systems in 479 genomes extracted from the biofilm metagenomes” (lines 24-26)? The authors used several artificial substrates which are no natural part of the Ocean to enrich unusual microorganisms. Do they expect endemic organisms on these artificial substrates or just substrates in general? These questions are getting more and more important and the authors should include the potential role of a “plastisphere” into their discussion. To go further, could artificial substrates even impact evolution and by this generate “artificial communities”? Taken together, a complex and new research field the authors are touching and the whole manuscript should be carefully revised concerning natural and artificial substrates. Both habitats have been investigated in the study, a more detailed comparison should be provided here.

Reply: The authors are very grateful for the positive feedback provided by the reviewer.

To clarify our statement regarding biosynthetic gene clusters and CRISPR-Cas, we have re revised the text (lines 28-30 in the marked-up manuscript) to: ‘Analysis of 479 genomes extracted from the biofilm metagenomes also revealed novel biosynthetic gene clusters and CRISPR-Cas systems.’

The reviewer raises a very pertinent point regarding the influence of substrate origin on biofilm composition. The biofilms in our study were developed on several artificial panels, including aluminum, poly(ether-ether-ketone), polytetrafluoroethylene, poly(vinyl chloride), stainless steel, titanium, and zinc. To examine the extent to which these ‘artificial communities’ contributed to the specificity of the biofilms, we undertook an additional principle coordinate analysis, comparing the similarity of biofilms developed on artificial panels with those developed on natural rocks, as well as comparing with seawater microbial communities. All samples included for this analysis were collected in Hong Kong seawater in 2017. Results indicate larger dissimilarity between biofilms on artificial panels and seawater samples than between biofilms on natural rocks and seawater samples (**Supplementary Figure 10**). Therefore, the ‘artificial communities’ are likely to contribute more to the discovery of novel functions than biofilms on natural substrates. These results are now

included in the SOM and we have included additional text in the main manuscript to address this point, with specific reference to biofilm colonization of plastics which are highly abundant in the global oceans (lines 164-171 in the marked-up manuscript).

Supplementary Figure 10. Similarity of the biofilms developed on artificial panels and natural rocks, as well as seawater microbial communities. All samples included for this analysis were collected in Hong Kong water in 2017. Jaccard similarity derived from the OTU matrix was illustrated by PCoA. These analyses were performed after normalizing the different 16S miTag data to the same library size, i.e. 10,000 sequences per sample.

Minor comments:

Line 71: Please mention explicitly that 16S rRNA genes are meant.

Reply: We have revised the text to state ‘16S rRNA gene sequences were extracted from the metagenomes for analysis (subsequently referred to as 16S miTags)’.

-Weipeng ZHANG

Reviewer #2 (Remarks to the Author):

The manuscript NCOMMS-18-28752 submitted by Zhang et al, A bank of undetected ocean microbial diversity and functional potentia, presents an in-depth look at a dataset that greatly expands our view of microbial marine life. The analyses appear to be well executed and appropriate to the description of the data. I see no obvious flaws in the work, and I consider it a significant addition to our understanding of the ocean.

Reply: The authors are very grateful for the positive feedback provided by the reviewer.

There is not enough detail given in the methods. In general, please provide ALL runtime parameters for software used, even if it's in the supplement. Two specific cases stood out to me:

In the Methods: Metagenome Analysis, you state: “[L]ow quality (quality score below 20) reads . . . were removed.” Quality scores are usually given per base not per read. This could be interpreted as an average score or a minimum score, please clarify. (Based on the software defaults, I'd guess it should be: quality below 20 for more than 30% of the read)

Reply: The runtime parameters for all software have now been provided in the manuscript text (lines 363-551 in the marked-up manuscript). Also, we amended the sentence “Low quality (quality score below 20) reads . . . were removed” to: “quality score <20 for >30% of the read’. Lines 373-374 in the marked-up manuscript.

In the Methods: Identification and annotation of the biofilm functional core gene set, cd-hit appears to be used twice. The first time, a 95% identity cutoff is mentioned, but no parameters are given for the second. Please provide all parameters used, not just the identity. There should have been some match length restrictions applied.

Reply: A cutoff of 95% sequence identity over 60% of the length of the shorter sequences was used, and this information has now been added to the manuscript text. Lines 464-491 in the marked-up manuscript.

My initial major complaint with the paper partially melted away when I re-read it just now. It was not clear to me at first if the paper claimed that it was the first comparison of metagenomes between planktonic and attached marine samples or if it is merely the first study to do it at scale. On closer inspection the middle paragraph of the introduction does cite previous work, but the final paragraph (taken on its own) may overstate things a bit. Clarifying the language would help readers understand the impact of the work. It think it is important to get it right in the introduction in order to properly put into perspective the later claims of the potential trove of data in these metagenomes.

Reply: This is certainly not the first oceanic comparison between biofilm-forming and free-living microbes. For instance, in the study by Brayant et al. (Diversity and activity of communities inhabiting plastic debris in the North Pacific Gyre. MSystems 2016.e00024-16), biofilms that formed on marine debris were compared with microbes in seawater, and different community structures were observed. However, these insights were primarily restricted to a localised scale. Our study is the first analysis to show the extent to which biofilm-forming microbes on different surface types contribute to global marine microbial diversity. We have revised the introduction so that this is clear and prior work is appropriately acknowledged. Lines 55-65 in the marked-up manuscript.

Two typos stood out:

CheckM is misspelled as ChechM.

Reply: Revised.

Citations 5 and 11 look like the same paper. Is that a duplicate or missing citation?

Reply: The citations were inadvertently duplicated and this has been corrected in the revised manuscript.

With the addition of all software parameters and clarification of the impact, I would recommend this paper for publication.

-J M Eppley

Reply: We have added the software parameters to the Methods section and have carefully revised the manuscript according to all other reviewer comments.

The authors wish more communications with you regarding microbial ecology study.

-Weipeng ZHANG

Reviewer #3 (Remarks to the Author):

Zhang and collaborators present a collection of metagenomic and 16S amplicon datasets from surface-attached and free-living communities in geographically distributed marine locations. The data collection reported here is a valuable resource for the scientific community. However, I have major reservations on the accompanying manuscript and analyses, as detailed next. Note that the list below is intended to highlight the clearest examples of each issue, but is not meant to be comprehensive.

Major Concerns

1. In my opinion, the introduction of this manuscript misrepresents the current state of the art and fails to justify the present study.

1a. As I state above, I believe the data presented in this study are a valuable contribution to the scientific community, but I fail to recognize this value out of the introduction. The aims of the study are presented in L58-62, and their justification in L55-57. However, the ecologic “principle” cited is actually a highly contested (if often cited) conjecture, and no other justification is given.

1b. Additionally, the introduction of the current manuscript would give the (erroneous) impression that this is the first report of this kind, and the first attempt at answering the posited aims. I understand the need for brevity, but failing to mention at the very least the (very similar) study by Bryant et al (mSystems 1, 2016) is a major omission of this section. In fact, that manuscript and references within (refs. 27-31 in Bryant et al 2016) clearly demonstrate that, indeed, the populations that form biofilms on plastics are distinct from those in the water column.

Reply: The authors are very grateful for the reviewer’s constructive comments which have considerably improved the manuscript. The innovative work by Bryant et al. (Diversity and activity of communities inhabiting plastic debris in the North Pacific Gyre), Zettler et al. (Life in the “plastisphere”: microbial communities on plastic marine debris), and Lobelle D et al. (Early microbial biofilm formation on marine plastic debris) focus on biofilms that formed on marine debris. The reviewer correctly notes that these studies addressed differences between biofilm-associated and free-living communities, however our study explicitly tests the contribution of biofilm communities to the global microbial diversity of the oceans. As noted above in response to both other reviewers, we have now modified the introduction to clearly reference this previous work, clarify the intent of our study and highlight the novelty of our findings. Lines 55-65 in the marked-up manuscript.

2. The inconsistency between results from taxonomic assignments of metagenome-derived 16S reads and 16S amplicons are understated.

2a. The authors recognize the differences between the two methods as presented in Suppl. Fig. 6, in comparison to Fig. 1c (L94-97), but these are simply discarded as an artifact of 16S copy number. No doubt this plays an important role in the difference, but its relative effect is never tested. Methods to account for copy-number variation exist (e.g., see Guo et al, AEM 82, Oct 2015), and other biases that could be affecting the analyses in this manuscript could play a role too.

Reply: We believe the reviewer is referring to the inconsistency between results of metagenome-derived 16S rRNA genes (miTags) and protein-coding maker genes. To address this in the revised manuscript, we have now added the taxonomic profile constructed using 16S miTags after gene length and copy number normalization. Such a normalization was performed using the software CopyRighter (Angly FE et al. 2014. *CopyRighter: a rapid tool for improving the accuracy of microbial community profiles through lineage-specific gene copy number correction. Microbiome, 2, 11*). The uncorrected taxonomic profile based on 16S

miTags (**Fig. 1c**), the taxonomic profile based on protein-coding marker genes (**Supplementary Fig. 6**) and the corrected community taxonomic profile based on 16S miTags (**Supplementary Fig. 7**) are also shown below. The relative abundance of most phyla using marker genes is more consistent with that of normalized 16S miTags, than with non-normalized 16S miTags (**Fig. 1C** and **Supplementary Fig. 7**). We also compared the relative abundance of Acidobacteria, Chloroflexi, Deltaproteobacteria, and Planctomycetes, which are enriched in biofilms compared with seawater (**Supplementary Fig. 8**), however we still see inconsistency between results based on 16S miTags (regardless of copy number normalization) and results based on marker genes, confirming that the discrepancy is only partially explained by variation in gene copy number and gene length. However, as illustrated in a recent study (*Louca, S. et al. 2018. Correcting for 16S rRNA gene copy numbers in microbiome surveys remains an unsolved problem. Microbiome, 6, 41*), 16S rRNA gene copy numbers can only be accurately predicted for a limited fraction of taxa, (i.e. those taxa with closely to moderately related representatives ($\leq 15\%$ divergence in the 16S rRNA genes)), hence correcting for 16S rRNA gene copy numbers in microbiome surveys remains largely unresolved. The biases in predicting community structure could be attributed to the incompleteness of the 16S rRNA gene copy number and protein-coding marker gene databases, which lack information for a large number of uncultured microbes. Since the central discovery of the present study is the uniqueness of the biofilm-derived microbes, rather than their relative abundance, we have added these results and discussion into the revised manuscript (lines 127-138 in the marked-up manuscript) but are unable to unequivocally resolve the discrepancy.

Figure 1c. Taxonomic composition based on analysis of metagenome-derived 16S rRNA genes (miTags) without gene length and copy number normalization. Abundant phyla (the top 30 phyla in terms of maximum relative abundance) are shown with all other phyla grouped together as ‘Minor’.

Supplementary Figure 6. Taxonomic composition based on protein-coding marker genes derived from metagenomes. The figure shows the distribution of all phyla across the 101 biofilm and 24 seawater metagenomes that were sequenced as part of the current study and the 67 previously published *Tara Oceans* metagenomes. The order of samples in this figure is the same as that in Supplementary Table 1.

Supplementary Figure 7. Taxonomic composition based on analysis of 16S miTags following gene length and copy number normalization. Abundant phyla (the top 30 phyla in terms of maximum relative abundance) are shown with all other phyla grouped together as ‘Minor’. The order of samples in this figure is the same as that in Supplementary Table 1.

Supplementary Figure 8. Representative microbial phyla differentially enriched in the biofilms (101 biofilms compared with 91 seawater samples). Abundance was calculated by analyzing 16S miTags, 16S miTags normalized by copy number and gene length, and protein-coding marker genes. ***P < 0.001 (Mann-Whitney U test).

3. The inconsistency between results from 16S amplicons and metagenome-derived 16S reads is not properly described or discussed.

3a. In L108-111, the authors indicate that the differences between taxonomic assignments from biofilm and water column communities are consistent across methods. This is not evident from Suppl. Fig. 8c v Suppl. Fig. 8d. In fact, the results indicate strong biases that are never discussed. Notably, the identification of biofilm-specific taxa is one of the main aims of this study, and yet the authors fail to identify the clear differences between the distributions presented in Suppl. Fig. 8a vs Suppl. Fig. 8b. 34% of the metagenome-derived 16S OTUs detected in biofilms are also detected in the water column, but this fraction is only 3% in the 16S amplicons. This indicates a strong bias in one or both methods that must be addressed. All analyses derived from these OTU assignments are invalidated by the possibility of such a large bias.

3b. Another clear example of these biases resides on the richness estimations. For example, the authors report a total of 25,379 OTUs detected in all 101 biofilm samples, and infer (based on rarefaction analysis) that this number is very close to saturation (L83-85). However, this figure is nearly matched by a single 16S amplicon library (Suppl. Fig. 9), and the total number of OTUs detected in only 9 amplicon libraries from biofilms was 58,615 (Suppl. Fig. 8a). In other words, less than 10% of the samples already show more than twice that number of OTUs, indicating that the sampling is actually very far from saturation. Using only the corresponding 9 metagenomic samples, the authors find only 8,038 OTUs, less than 15% of those observed in the amplicons. Similarly, in L116-117 the authors indicate that the median richness in biofilms was 2,790, about 10 times less than the values shown in Suppl. Fig. 9. These results clearly contradict the statement that “microbial diversity was adequately captured” and further underscore the issues raised in point #2.

Reply: We appreciate the reviewers point regarding inconsistency between results obtained using 16S rRNA gene amplicon analysis with those from our metagenome-derived 16S reads. Likely reasons for the discrepancy between approaches include sequencing depth and the different 16S rRNA gene regions. We sequenced ~10-20 Gb datasets per metagenome, generating about 10-30,000 16S rRNA gene sequences (miTags) per sample. In contrast, a single amplicon analysis generated >100,000 16S rRNA gene sequences per sample. Moreover, the 16S rRNA genes extracted from metagenomes correspond to a variety of regions from the 16S molecule, while amplicon analysis exclusively targeted the V3-V4 region. Analysis of the 16S miTags from 101 biofilms revealed that 7,300 seawater-derived biofilm-forming ‘species’ were undetected in seawater, while analysis of nine 16S amplicon samples revealed even more biofilm-specific OTUs. This would indeed suggest that an even greater number of species inhabiting biofilms are not present in seawater, and thus the global microbial diversity has been underestimated. This is entirely consistent with the main conclusions of our study. However, to avoid any unnecessary confusion regarding the different approaches, we have decided to omit the amplicon analysis from the results.

To test the effect of sequencing depth on the discovery of biofilm-specific OTUs, we sequenced two biofilm samples and two seawater samples collected from the same site at the same time to generate more than 110 gigabytes of data per sample and calculated the biofilm and seawater specificities. Based on extraction of 20, 40, 80, and 160 million reads, the specificity of the biofilms (the ratio of biofilm-specific OTUs to the total number of OTUs in biofilms and seawater) increased slightly with sequencing depth, following a linear relationship, whereas that of the seawater sample decreased with sequencing depth (**Supplementary Fig. 5**). Based on 160 million reads, 33.1% of the biofilm OTUs could not be detected in seawater. Therefore, if we increased sequencing depth further, it is likely that the number of biofilm-specific species would increase even higher than the 7,300 stated in the current version of manuscript. Hence, the reviewer is correct that the microbial diversity in the biofilms has not been fully captured, and we have deleted this statement from the

text accordingly (lines 101-114 in the marked-up manuscript). Importantly however, this result only further confirms our key conclusion that marine biofilms host exceptional and unique microbial diversity not detected in the surrounding seawater, and thus the global microbial diversity of the oceans has been grossly underestimated. The rarefaction curve is a gamma-diversity (as pointed out by the reviewer).

Supplementary Figure 5. Correlation between biofilm or seawater specificity and sequencing depth. A series of numbers (20, 40, 80, and 160 million) of reads were randomly extracted, from one deeply sequenced biofilm metagenome and one seawater metagenome. a) The number of OTUs specific to biofilms or seawater. b) The ratio of specific OTUs in biofilms or seawater compared to the total number of OTUs in biofilms and seawater.

4. The assumptions of several key analyses are violated or left untested.

4a. The analysis in Suppl. Fig. 4 was performed on number of samples, not on sample size, and is therefore an indication for gamma diversity, not alpha diversity, but this is not clearly indicated in the text. More importantly, this analysis was performed on classified OTUs, and is therefore biased by the incompleteness of the databases. Although the authors don't report the degree of novelty of the datasets presented (only the differences with respect to the TARA collection), presumably these datasets are not fully captured by existing sequences in the tested databases. This violates the assumptions behind collector curves.

4b. L325-354: "... only OTUs identified in both forward and reverse paired reads in a sample were retained (thus all singletons were removed)". Does this mean that each pair was counted twice? If yes, several statistical analyses that assume independence of observations are compromised. If not, and the authors meant to indicate that, in addition, singletons were removed, how were the Chao1 estimators of richness calculated? Note that Chao1 uses the number of OTUs seen exactly once (singletons).

4c. L111-113 identifies 500,000 as the number of reads per sample necessary to saturate sequencing, and yet

Fig. 3 presents analyses performed on rarefied datasets at only 10,000 sequences per sample. In fact, rarefying samples is not a robust normalization method and should be avoided altogether (e.g., see McMurdie and Holmes, PLoS CB 10(4), 2014). Also, see minor issue #5.

4d. Suppl. Fig. 12 shows that the most abundant category of resistance (after “Unclassified”) was “RequiresSNPCconfirmation_1”. No details are provided in the methods or the legend, so it would appear this simply corresponds to genes that may confer resistance with specific SNPs. Was this further examined? These genes may correspond to housekeeping genes, potentially inflating the statistics presented by the authors on antibiotic resistance.

Reply: The reviewer is correct that we present gamma diversity, not alpha diversity. We used this analysis to show that the number of samples was appropriate to study the diversity of microbes in marine biofilms. However, as indicated in our sequencing depth analysis (**Supplementary Fig. 5**), a greater number of biofilm-specific species could be detected with increasing sequencing depth for a given metagenome. Given that we already sequenced more than 2 terabases of datasets, it will be challenging to capture the entire microbial diversity present in marine biofilms. In addition to the newly sequenced 24 seawater samples, we also used the Tara Oceans data sets as a reference because this is the largest existing microbial dataset from the marine environment.

‘Only OTUs identified in both forward and reverse paired reads in a sample were retained (thus all singletons were removed)’. We undertook this approach to analyze the taxonomic structure of the biofilm and seawater communities. When we prepared the matrix for alpha diversity (including Chao 1) and beta diversity analyses, we used only the forward reads, and thus singletons were included. We have revised this section of the text accordingly (lines 451-462 in the marked-up manuscript).

Regarding the query about rarefying samples, in the revised manuscript we also performed PCoA using the total miTags extracted from the forward reads of the metagenomes (101 biofilms and 91 seawater samples), rather than only using 10,000 miTags per sample. The PCoA figure is shown in **Supplementary Figure 9**. The clear differentiation between the biofilms and seawater samples is still evident and this additional figure and associated discussion has been inserted in the revised manuscript (lines 157-164).

A single-nucleotide polymorphism-based cluster grouping classification system for *Mycobacterium tuberculosis* was used to examine antibiotic resistance type (*Brimacombe et al. 2007. Antibiotic resistance and single-nucleotide polymorphism cluster grouping type in a multinational sample of resistant Mycobacterium tuberculosis isolates. Antimicrobial agents and chemotherapy, 51, 4157-4159*). However, there is no clear definition for the category ‘RequiresSNPCconfirmation_1’, and thus the type of antibiotic it resists remains unclear. To avoid unnecessary confusion we have placed this in an ‘Unclassified’ category within Supplementary Fig. 13 of the revised manuscript.

Supplementary Figure 9. Similarity of the microbial communities based on 16S miTags without data size normalization. Jaccard distances were calculated based on an OTU matrix derived from 16S miTags without data size normalization, and visualized through PCoA.

Supplementary Figure 13. Antibiotic resistance genes identified in the BCGC. MLS refers to macrolides, lincosamides, and tetracyclines.

Minor Issues

1. The manuscript creates a good deal of unnecessary confusion by avoiding explicit labeling of 16S amplicons vs miTags. A reader can find all the information with some effort, but clearly stating the base data of each analysis could simplify the reading.

Reply: As stated above, to avoid any unnecessary confusion we have decided to remove the 16S rRNA gene amplicon analysis from the manuscript.

2. The first mention of miTags (L79) is not accompanied by an explanation of what it is.

Reply: We have added an explanation for the miTags (lines 86-87 in the marked-up manuscript): “16S rRNA genes derived from Illumina-sequenced environmental metagenomes”.

3. L71: “...resulted in a total of 13,064 unique OTUs in the 101 biofilms...”. This phrasing can introduce some confusion, since many readers could interpret “unique OTUs” as total OTUs after dereplication, as opposed to the (intended) meaning of “OTUs unique to biofilms”.

Reply: We have revised to “OTUs unique to biofilms”.

4. L88-89: “... excluding any impact from DNA extraction bias on community structure of the biofilms”. Not really. In fact, Suppl. Fig. 5 clearly shows that the biases introduced by different methods could be sample-dependent. This means that a much larger number of samples would be necessary to exclude any impact. In addition, the same figure shows that the DN2 has a larger dissimilarity to TIAN2 and A1IP2 than these exhibit to sample 1 (with any extraction method). Since there is no ground truth dataset, and the variability introduced by the methods doesn't appear to be systematic (at least based on those two samples), there is an important probability that the TIANamp method (used in the rest of the study) has a strong impact on (observed) community structure.

Reply: The reviewer raises a valid point. Importantly however, the DNA of the vast majority of biofilm samples and all newly generated seawater samples was extracted using the same kit, hence are directly comparable. As we did not conduct a systematic analysis of all extraction kits using all biofilm substrates (as this was not the focus of the study), we have opted to remove this section from the manuscript.

5. The authors indicate in L11-113 that saturation is reached at 500,000 reads per sample, but the horizontal axis of Suppl. Fig. 9 only extends to 410,000.

Reply: This figure, as well as the 16S amplicon results text has been removed from the revised manuscript.

6. L244-245: “We surmise that the biofilm core gene set underpins environmental adaptation and microbe-microbe interaction”. It would be valuable to further test other indicators of interaction such as toxins/antitoxins or Quorum-sensing signals.

Reply: The reviewer raises an interesting point and while we attempted this analysis, we did not detect Quorum sensing genes in the biofilm core gene catalogue. Based on the annotation by the Pfam database, we found several EAL domain-containing genes (PF00563) which may relate to signal transduction through cyclic AMP, a molecule that modulates Quorum sensing in a number of bacterial species (Liang W et al. The cyclic AMP receptor protein modulates quorum sensing, motility and multiple genes that affect intestinal colonization in

Vibrio cholerae. Microbiology, 153, 2964-2975).

7. L88-89 and Suppl. Fig. 5: should be “observed” community structure and taxonomic composition. Neither actual community structure nor the actual taxonomic composition are affected by DNA extraction.

Reply: These results have been removed from the revised manuscript

8. L389: USEARCH.

Reply: Revised accordingly

9. L456: “... the Software rRNA prediction”, presumably refers to Meta_RNA.

Reply: Yes. This is Meta_RNA and this has been inserted in the revised manuscript (line 513 in the marked-up manuscript).

10. L460-461: “using the maximum likelihood as the substitution model”. Please review the methods carefully, this is a major error indicating that this section was not written carefully, and could give the reader the impression that it wasn’t performed very carefully either.

Signed: Luis M. Rodriguez-R.

Reply: This section has been revised appropriately (lines 516-518 in the marked-up manuscript): “The maximum-likelihood phylogenetic tree was constructed in MEGA (version 7.0) using the Tamura-Nei substitution model and bootstrapping was performed with 1,000 repetitions.”

I read through your publications (e.g., How much do rRNA gene surveys underestimate extant bacterial diversity?) and found these analyses are very helpful for my future research. I wish more communications with you in future to improve my research ability.

-Weipeng ZHANG

REVIEWERS' COMMENTS:

Reviewer #1 (Remarks to the Author):

Review on Zhang et al., 2018

Title: Biofilms constitute a bank of hidden microbial diversity and functional potential in the oceans

Authors: Weipeng Zhang, Wei Ding, Yong-Xin Li, Chunkit Tam, Salim Bougouffa, Ruojun Wang, Bite Pei, Hoyin Chiang, Pokman Leung, Yanhong Lu, Jin Sun, He Fu, Vladimir B Bajic, Hongbin Liu, Nicole S Webster, Pei-Yuan Qian

As already mentioned in the review of the first version, aim of Zhang et al. was to increase knowledge on the microbial diversity in the Ocean, on taxonomical and functional level. The approach of the authors was to enrich not yet detected members of the seawater communities by stimulating their growth on artificial surfaces in the Ocean (like PVC etc.). Eventually, the authors investigated the microbial composition by generating and analyzing 2.3 terabases sequenced datasets (metagenomes but also more specific 16S rRNA gene analyses) and in principal compared it to the Tara Oceans metagenomes. Alpha- and beta-diversity was determined on the biofilm and seawater metagenomes; moreover, the biofilm functional genes were identified and binned. As result, more than 7,300 formerly unknown 16S rRNA gene OTUs were detected in the biofilms. Novel lineages include, for instance, the Acidobacteria. They must have derived from seawater and probably belonged to its rare organism fraction which often remains undetected by currently established methods. Zhang et al. expected that the known microbial diversity increased by more than 20% using their biofilm approach. They also identified a functional core across the biofilms likely play a role in stress responses and microbe-microbe interactions.

As stated in my first review, some conclusions made by the authors remained unclear to me. However, these concerns have sufficiently addressed by the authors in the revised version, for instance by providing suppl. Fig. 10, demonstrating the dissimilarities between biofilms on rocks and artificial substrates in more detail. Thus, I suggest acceptance of the manuscript now.
Matthias Labrenz

Reviewer #2 (Remarks to the Author):

In their revised manuscript "Biofilms constitute a bank of hidden microbial diversity and functional potential in the oceans", Zhang et al have sufficiently addressed my concerns.

Signed: J M Eppley

Reviewer #3 (Remarks to the Author):

The authors present here a much-improved version of the manuscript. In the current version they have either addressed my concerns or removed the sections or conclusions that were not supported or appeared to be flawed. I particularly celebrate the additional context provided in the introduction and the additional experiment with deeply sequenced samples from adjacent biofilm and seawater samples.

Unfortunately the authors decided to completely remove the amplicon data, which provided an

interesting contrast that could have been further explored. I'm looking forward for a follow-up study on this issue.

On a very minor note, there is a typo in line 460: miTagas should be miTags.

Signed: Luis M Rodriguez-R

REVIEWERS' COMMENTS:

Reviewer #1 (Remarks to the Author):

Review on Zhang et al., 2018

Title: Biofilms constitute a bank of hidden microbial diversity and functional potential in the oceans
Authors: Weipeng Zhang, Wei Ding, Yong-Xin Li, Chunkit Tam, Salim Bougouffa, Ruojun Wang, Bite Pei, Hoyin Chiang, Pokman Leung, Yanhong Lu, Jin Sun, He Fu, Vladimir B Bajic, Hongbin Liu, Nicole S Webster, Pei-Yuan Qian

As already mentioned in the review of the first version, aim of Zhang et al. was to increase knowledge on the microbial diversity in the Ocean, on taxonomical and functional level. The approach of the authors was to enrich not yet detected members of the seawater communities by stimulating their growth on artificial surfaces in the Ocean (like PVC etc.). Eventually, the authors investigated the microbial composition by generating and analyzing 2.3 terabases sequenced datasets (metagenomes but also more specific 16S rRNA gene analyses) and in principal compared it to the Tara Oceans metagenomes. Alpha- and beta-diversity was determined on the biofilm and seawater metagenomes; moreover, the biofilm functional genes were identified and binned. As result, more than 7,300 formerly unknown 16S rRNA gene OTUs were detected in the biofilms. Novel lineages include, for instance, the Acidobacteria. They must have derived from seawater and probably belonged to its rare organism fraction which often remains undetected by currently established methods. Zhang et al. expected that the known microbial diversity increased by more than 20% using their biofilm approach. They also identified a functional core across the biofilms likely play a role in stress responses and microbe-microbe interactions. As stated in my first review, some conclusions made by the authors remained unclear to me. However, these concerns have sufficiently addressed by the authors in the revised version, for instance by providing suppl. Fig. 10, demonstrating the dissimilarities between biofilms on rocks and artificial substrates in more detail. Thus, I suggest acceptance of the manuscript now.

Matthias Labrenz

Reply: The authors are grateful for your very positive feedback.

Weipeng Zhang

Reviewer #2 (Remarks to the Author):

In their revised manuscript "Biofilms constitute a bank of hidden microbial diversity and functional potential in the oceans", Zhang et al have sufficiently addressed my concerns.

Signed: J M Eppley

Reply: The authors are grateful for your very positive feedback.

Weipeng Zhang

Reviewer #3 (Remarks to the Author):

The authors present here a much-improved version of the manuscript. In the current version they have either addressed my concerns or removed the sections or conclusions that were not supported or appeared to be flawed. I particularly celebrate the additional context provided in the introduction and the additional experiment with deeply sequenced samples from adjacent biofilm and seawater samples. Unfortunately the authors decided to completely remove the amplicon data, which

provided an interesting contrast that could have been further explored. I'm looking forward for a follow-up study on this issue. On a very minor note, there is a typo in line 460: miTagas should be miTags.

Signed: Luis M Rodriguez-R

Reply: The authors are grateful for your very positive feedback. The typo has been corrected. We will pay attention to the difference between 16S amplicon data and metagenomes in our future studies.

Weipeng Zhang